# TOPOFORMER: BRAIN-LIKE TOPOGRAPHIC ORGANIZATION IN TRANSFORMER LANGUAGE MODELS THROUGH SPATIAL QUERYING AND REWEIGHTING

**Taha BinHuraib**
Novus Technologies
taha@novuswriter.com

**Greta Tuckute**
MIT
gretatu@mit.edu

**Nicholas M. Blauch**
Harvard University
nblauch@fas.harvard.edu

## ABSTRACT

Spatial functional organization is a hallmark of biological brains: neurons are arranged topographically according to their response properties, at multiple scales. In contrast, representations within most machine learning models lack spatial biases, instead manifesting as disorganized vector spaces that are difficult to visualize and interpret. Here, we propose a novel form of self-attention that turns Transformers into "Topoformers" with topographic organization. We introduce *spatial querying* — where keys and queries are arranged on 2D grids, and local pools of queries are associated with a given key — and *spatial reweighting*, where we convert the standard fully connected layer of self-attention into a locally connected layer. We first demonstrate the feasibility of our approach by training a 1-layer Topoformer on a sentiment classification task. Training with spatial querying encourages topographic organization in the queries and keys, and spatial reweighting separately encourages topographic organization in the values and self-attention outputs. We then apply the Topoformer motifs at scale, training a BERT architecture with a masked language modeling objective. We find that the topographic variant performs on par with a non-topographic control model on NLP benchmarks, yet produces interpretable topographic organization as evaluated via eight linguistic test suites. Finally, analyzing an fMRI dataset of human brain responses to a large set of naturalistic sentences, we demonstrate alignment between low-dimensional topographic variability in the Topoformer model and human brain language network. Scaling up Topoformers further holds promise for greater interpretability in NLP research, and for more accurate models of the organization of linguistic information in the human brain.

## 1 INTRODUCTION

Biological brains are spatially organized, containing category-selective areas (Kanwisher, 2010), broad feature maps that tile individual cortical areas (Konkle & Oliva, 2012; Bao et al., 2020) and the cortex more broadly (Huth et al., 2012; 2016; Margulies et al., 2016), as well as large-scale distributed networks (Yeo et al., 2011; Braga et al., 2020). This spatial organization is one way in which the human brain, a vastly complex "black box", is more naïvely interpretable than modern deep neural networks (DNNs), whose units have functional properties organized without simple spatial priors. Recent work in computational neuroscience has bridged this gap in DNNs trained for vision, demonstrating that local smoothness or wiring cost minimization objectives can be incorporated into DNNs to encourage the development of smooth functional organization of responses, which can then be easily visualized in 2D (Lee et al., 2020a; Blauch et al., 2022; Keller & Welling, 2021; Doshi & Konkle, 2021; Margalit et al., 2023; Lu et al., 2023), building upon classic approaches (Kohonen, 1982; Jacobs & Jordan, 1992). In addition to simulating topographic properties within regions, topographic vision models have also explained the hierarchical organization of topographic information from earlier to later visual areas (Margalit et al., 2023; Lu et al., 2023). One topographic vision model has even demonstrated the emergence of spatial clusters corresponding to ventral, lateral, and dorsal streams of the visual system (Finzi et al., 2021; 2023). Collectively, topographic

vision models are helping to unify a computational understanding of the functional organization of the visual system.

However, topographical priors have not yet been built into models of linguistic processing, despite tremendous progress in the development of natural language processing (NLP) models and their application in cognitive science and neuroscience (Wilcox et al., 2020; Gauthier et al., 2020; Schrimpf et al., 2021; Caucheteux & King, 2022; Goldstein et al., 2022a; Tuckute et al., 2024). In NLP, Transformer language models (LMs) have undoubtedly established themselves as the leading architecture for language tasks (Vaswani et al., 2017; Radford et al., 2018; Brown et al., 2020; OpenAI, 2023), displaying human-like language understanding and generation. In cognitive science and neuroscience, these LMs have emerged as the most quantitatively accurate models of human language processing. They generate probabilities of upcoming words that explain reading behavior of humans (Wilcox et al., 2020; Merkx & Frank, 2021; Shain et al., 2022; Oh & Schuler, 2023), and their internal activations can explain the neural signals of humans reading or listening to naturalistic sentences or stories at the granularity of fMRI voxels and intracranial recordings (Schrimpf et al., 2021; Goldstein et al., 2022b; Caucheteux & King, 2022; Antonello et al., 2024; Tuckute et al., 2024). Despite the success of these LMs, they remain difficult to interpret, and incomplete as models of brain function.

In the current work, our aim is to bridge these gaps by inducing a topographic organization of features within the Transformer architecture. We employ local-connectivity based approaches inspired by recent topographic vision models (Keller & Welling, 2021; Blauch et al., 2022) to the language domain, asking whether we can obtain topographic organization of linguistic representations within a Transformer architecture via spatial constraints. To do so, we introduce two computational motifs — *spatial querying* and *spatial reweighting* — to the self-attention layer, which encourage the development of topographic organization in separate components of the self-attention layer. We call Transformer models employing these constraints **Topoformers**. We show that we can scale these topographic motifs to a large BERT Topoformer model trained with a masked language modeling objective, and that topographic organization develops within each hierarchical layer of the network, without significantly compromising task performance. We interpret this topography using a novel suite of 8 semantic and syntactic tests. Last, we demonstrate that the topographic representations of the Topoformer can be aligned with the topographic representations of the human functionally-defined language network in multiple subjects. In summary, our work demonstrates for the first time that Transformer models can be trained to exhibit topographic organization similar to the human brain, and paves the way for further interpretability work leveraging spatial priors.

## 2 METHODS

In this study, we propose two approaches for enforcing topographic organization in a Transformer layer. Both methods rely on the use of local communication to introduce spatial constraints that encourage the formation of spatially organized linguistic representations.

### 2.1 SPATIAL QUERYING

We begin with the standard self-attention operation used by Vaswani et al. (2017). In this formulation, every token embedding is projected onto a set of queries, keys, and values, and the query of a given token is associated with a corresponding key of all other tokens. Spatial querying works by associating a local pool of queries with a given key. The locality is parameterized with a width parameter $r_{SQ}$ determining the fraction of units in a given key's circular receptive field (RF). For simplicity, we examine the case of a simple non-weighted sum of queries. This is achieved by inserting a binary intermediate matrix $M \in \mathbb{R}^{d \times d}$, where $d$ is the embedding dimension, and the columns of M determine the spatial pool of queries associating with a given key. Thus, the term $QK^T$ is replaced with $QMK^t$, such that the dot product attention between a given pair of tokens is not between individual queries and keys, but local pools of queries and individual keys (or equivalently, vice versa). This biases the representations of queries to be locally smooth, and the representations of keys to have a spatial correspondence with the queries. For an intuitive explanation of spatial querying, see Appendix A.1.

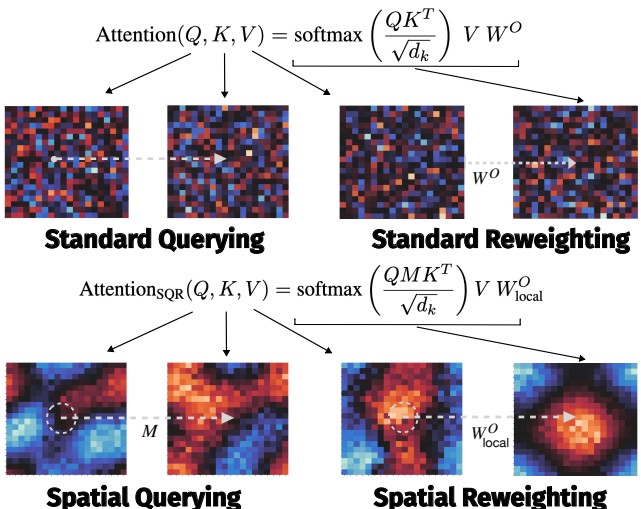

**Figure 1: Spatial querying and reweighting operations in the "Topoformer".**
The first row shows standard querying operations in the attention module of a single-head Transformer, and the second row shows the spatial counterparts used in the Topoformer. Standard querying associates a single query dimension of token $i$ with a single key dimension of token $j$. In contrast, spatial querying associates a local pool of query dimensions with a given key dimension, through the intermediate local pooling matrix M. Standard (dense) reweighting applies a fully connected linear layer $W^O$ to the outputs of a single attention head (typically to combine the outputs of multiple attention heads). In our formulation, we use a locally connected layer $W^O_{\text{local}}$ in its place (spatial reweighting). While the figure illustrates querying for a pair of tokens and reweighting for a single token, when processing a full sequence, there is a 2D grid of the form shown here for each token. Each heatmap shows the loadings of the second PC of responses (top: control model, bottom: Topoformer-SQR model with spatial querying and reweighting).

To ensure that the dominant functional organization occurs *within* a head rather than *across* heads, we use a single attention head in the Topoformer implementation, but we retain the outer reweighting matrix $W^O$ used in multi-head attention (Vaswani et al., 2017). Without further constraints than Eq 2, organization across heads would be non-topographic and thus complicate interpretability and visualizations. For a visual explanation, Figure 1 compares the differences between standard operations within a self-attention block, and their spatial counterparts. While we work with square grids of dimensionality $d = s^2$, where $s$ is the grid side length, theoretically, any 1D, 2D, or 3D arrangements of units could be used to define the spatial position of units.

## 2.2 SPATIAL REWEIGHTING

Spatial querying only imposes a topographic relationship between queries and keys; to encourage the development of topographic organization of the values and self-attention outputs (hereafter fc_out), we convert the outer reweighting matrix $W^O$ to a locally connected layer $W^O_{local}$. By using locally connectivity in $W^O_{local}$, we encourage the model to learn more localized feature representations in the values and attention outputs. We parameterize local connectivity using a width parameter $r_{SR}$ that determines the fraction of units within a given unit's circular receptive field (RF). Putting spatial querying and reweighting together, we arrive at the final modified self-attention equation:

$$\text{Attention}_{\text{SQR}}(Q, K, V) = \text{softmax}\left(\frac{QMK^T}{\sqrt{d_k}}\right) V \, W^O_{\text{local}} \tag{1}$$

Preliminary experiments demonstrated the need to use large positive weights to fully encourage the development of topographic organization. Thus, we initialize $W^{O+}_{local} = |W^O_{local} * 10|$, where $W^0_{local}$ is a standardly initialized locally connected layer. This operation, denoted as spatial reweighting, has the effect of enhancing local correlations, commonly viewed as a hallmark of topographic organization (Lee et al., 2020b; Blauch et al., 2022; Margalit et al., 2023). These excitatory feedforward

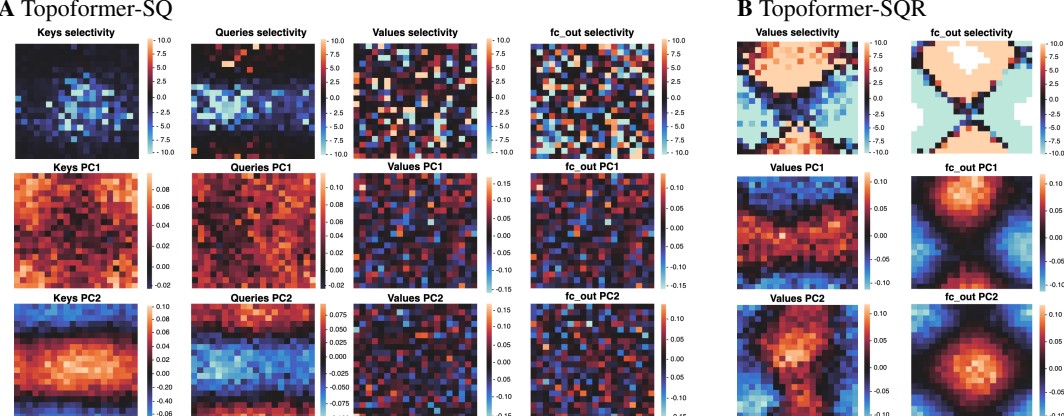

**Figure 2: Topographic organization across sublayers with spatial querying and reweighting.**
**A.** Topoformer-SQ produces topography in the keys and queries, but not the values or self-attention outputs. Each column shows a different sublayer representation within a self-attention block (keys, queries, values, and fc_out). The representations were obtained by averaging across the tokens in each sentence from the IMDB sentiment classification test set (Maas et al., 2011). The first row shows selectivity for positive vs. negative sentiment sentences. We plot the selectivity significance value (see Appendix A.3.2), where $s = 2$ corresponds to positive selectivity with $p = 0.01$, and $s = -2$ corresponds to negative selectivity with the same significance level. The second and third rows show the PC weights for the first and second components, respectively. **B.** Topoformer-SQR produces topography in the values and self-attention outputs. The format is the same as for panel A, but for brevity we show only the values and self-attention outputs as the keys and queries show a similar topographic organization from spatial querying.

connections mimic the dominant role of excitatory pyramidal neurons in between-area cortical communication in biological brains, and boostrap the effect of local connectivity on enhancing local correlation (Laszlo & Plaut, 2012; Blauch et al., 2022).

## 3 RESULTS

### 3.1 TRAINING A 1-LAYER TOPOFORMER ON A SUPERVISED TASK

We begin by training 1-layer single-head encoder-only Topoformer (bidirectional attention) on the IMDB sentiment analysis dataset (Maas et al., 2011), which classifies movie reviews as having a positive or negative overall sentiment. The local connectivity in the spatial querying and spatial reweighting operations are controlled through hyperparameters $r_{SQ}$ and $r_{SR}$, respectively, that sets the radius of spatial receptive fields (RFs). We investigated the effect of different RF sizes in a 1-layer Topoformer model with spatial querying and reweighting (Topoformer-SQR, Section 3.1.2), finding that smaller $r_{SQ}$ values yield better accuracy and topography, while the network is more robust to the $r_{SR}$ hyperparameter (see Appendix A.2 for details). In the following, we report results for an RF size of 0.3 for a model with just spatial querying (SQ; Section 3.1.1) and 0.1 for the model with spatial querying and reweighting (SQR; Section 3.1.2), with $r_{SQ} = r_{SR}$. We set $d = 400$.

### 3.1.1 TOPOFORMER-SQ

We first describe the results of a model using only spatial querying (**Topoformer-SQ**). Following training, Topoformer-SQ achieved an accuracy of 0.81 on the IMDB sentiment test set. In comparison, an identical 1-layer Transformer model without spatial querying achieved an accuracy of 0.83. To probe its topographic organization, we conducted selectivity analyses and principal component analysis (PCA) to investigate the unit activation patterns in the different layers, as shown in Figure 2 (see Appendix A.3, A.4 for details). Our selectivity analysis was designed to contrast the response magnitudes for positive and negative sentiment sentences. As expected, this analysis revealed a

topographic organization in the keys and query sublayers only. We next performed PCA to assess generic forms of topographic variability in the model representations. We found that the weights of the first two principal components (PCs) exhibited a smooth topography in the keys and queries, with the second PC spatially aligned to the selectivity for both representations. This demonstrates that the network has learned to organize its dominant modes of variability topographically.

### 3.1.2  TOPOFORMER-SQR

We next trained a model incorporating both spatial querying and reweighting (**Topoformer-SQR**). This model achieved a test set accuracy of 0.75 on the IMBD sentiment test set, slightly lower than the Topoformer-SQ model. We performed identical probing analyses to those in the previous section, highlighting the results for the values and fully-connected (fc_out) representations in Figure 2B. We found that the Topoformer-SQR exhibited more pronounced topographic organization in the values and fc_out layers compared to Topoformer-SQ. This observation suggests that the local connectivity matrix $W_{\text{local}}^{O}$ (see Figure 1A., and Equation 1) successfully enforced a topographic correspondence between the values and attention outputs, as predicted. We also quantified the extent of this topography by using a statistic $t_g$ that relates the degree of correlation with the distance between pairs of units (see Appendix A.4.2, Equation 4, Figure 9). A high value of $t_g$ indicates that nearby units tend to be more correlated in their response pattern across sequences than distant units.

### 3.2  SCALING UP: TOPOFORMER-BERT

| BERT Model | MNLI | SST-2 | STSB | RTE | QNLI | QQP | MRPC | CoLA | GLUE |
|---|---|---|---|---|---|---|---|---|---|
| multihead | 83.0/83.2 | 91.6 | 84.8 | 54.7 | 88.5 | 86.9 | 86.4 | 43.7 | 78.1 |
| 1 head | 81.1/81.5 | 90.0 | 82.1 | 51.2 | 87.6 | 86.7 | 84.8 | 47.5 | 76.9 |
| **Topoformer** | 80.1/80.1 | 90.9 | 75.1 | 51.2 | 86.6 | 86.0 | 81.5 | 46.3 | 75.31 |

**Table 1:** Comparison of GLUE performance between multi-head and single-head attention non-topographic BERT control models and Topoformer-BERT (single-head attention), each trained with the Cramming procedure (Geiping & Goldstein, 2022).

We next scaled up the Topoformer motifs to train a BERT model using a Masked Language Modeling objective (**Topoformer-BERT**). We followed the training paradigm introduced by (Geiping & Goldstein, 2022). We trained a 16-layer BERT model on the Bookcorpus-Wikipedia dataset (Zhu et al., 2015) for 12 hours (see Appendix A.6 for more details). To provide a control for our Topoformer-BERT model, we trained a standard, non-topographic single-head BERT model with identical parameters and training procedure as our Topoformer-BERT (besides the lack of topographical motifs, Appendix A.5). To evaluate the models' performance on natural language tasks, we followed the General Language Understanding Evaluation (GLUE) benchmark (Wang et al., 2019) procedure as described in (Geiping & Goldstein, 2022), testing each model on all tasks besides WNLI as in (Devlin et al., 2019). Critically, we observed that the task performance of Topoformer-BERT on the GLUE Benchmark was similar to that of the non-topographic model counterpart, suggesting that our added spatial constraints were not significantly hindering task performance (Table 1). Having established that Topoformer-BERT is capable of performing linguistic tasks, we move on to characterizing the topographic organization in Topoformer-BERT.

First, we systematically quantified the topography of each of the 16 Topoformer-BERT layers using the statistic described in Section 3.1.2. We plot the mean statistic $\bar{t}_g$ over a range of scales for all layers in Figure 3A, and the distance-threshold-specific statistic $t_{g,d}$ for layer 15 (Figure 3B). In general, the keys and queries have the greatest degree of topographic organization (also visually evident in Figure 3C), and the values show the weakest organization. Nevertheless, each is consistently above 0, typically driven by very local decay in correlation, as seen in the analysis of $t_{g,d}$ across different maximum distances (Figure 3).

Second, we took an initial step towards interpreting the emergent topographical structure in Topoformer-BERT. Specifically, we evaluated the selectivity of the unit activations to eight basic test suites targeting different linguistic properties. All eight test suites consisted of 76 sentences each, and were either based on carefully designed minimal pair sentences based on prior work (Gauthier et al.

**Figure 3: Topographic organization across all layers of Topoformer-BERT.**
**A.** We quantified the extent of topography in Topoformer-BERT via the generic topography statistic $\bar{t}_g$ (Equation 4) which intuitively relates the degree of correlation with the distance between pairs of units averaged over a range of scales (nine maximum distance values). **B.** The topography statistic $t_{g,d}$ for layer 15 computed at each of a range of scales (maximum distance value), highlighting the particularly strong local topography for queries and keys. **C.** Visualization of the first principal component (PC) weights for keys and fc_out sublayers.

(2020); Hu et al. (2020); Misra et al. (2023)) or were designed by us to control for the number of words and sentence surprisal (see Appendix A.7).

The first suite, **Intactness** tests intact sentences versus their scrambled counterparts, thereby degrading both linguistic form (syntax) and meaning (semantics). The next suites test more targeted semantic properties: Suites 2 through 4 test three different dimensions of *meaning* that have been extensively investigated in prior work, as specified below. Suite 2 tests **Animacy** (sentences with animate vs. inanimate meanings; Naselaris et al. 2009; Connolly et al. 2012; Konkle & Caramazza 2013), suite 3 tests **Concreteness** (sentences with concrete vs. abstract meanings; Binder et al. 2005; Fiebach & Friederici 2004), and suite 4 tests **Visuomotor** properties (sentences with visual vs. motor meanings; Desai et al. 2010; Lynott et al. 2020). The next suite (5) tests **Semantic acceptability** using minimal pair sentences ( Misra et al. 2023). The final three suites test three different dimensions of *form* using suites from SyntaxGym Gauthier et al. (2020); Hu et al. (2020): Suite 6 tests **Agreement** (Subject-Verb Number Agreement), suite 7 tests **Licensing** (Reflexive Number Agreement), and suite 8 tests **Garden-Path** ambiguity (Verb Transitivity).

We performed selectivity analyses for these eight test suites (Figure 4). These analyses intuitively ask whether a given unit shows a preference for a particular contrast (e.g., animate versus inanimate sentences). First, we evidenced a strong topographic organization according to linguistic content from different semantic categories (top row, Figure 4), both in terms of significant topographic selectivity, as well as significant decodability of condition from the distributed pattern of activities via a logistic regression classifier (Appendix A.6). Intriguingly, the selectivity patterns were different across contrasts, implying that semantic distinctions are represented in topographic activity pattern differences across categories. Second, we turned to more controlled test suites constructed using pairs of minimally different sentences in line with prior work in psycholinguistics and NLP (e.g., Linzen et al. 2016; Warstadt et al. 2020). As expected, the effects were lower (bottom row, Figure 4) relative to sentences only matched on length and surprisal (top row). We evidenced weak topographic selectivities to sentences with correct syntactic agreement versus those that do not, for example. The weak selectivities (e.g., Semantic acceptability in Figure 4) were also reflected in poorer decoding over the patterns of keys activation (chance-level 50%; in the case of Licensing (accuracy $\approx$ .18), the use of a Ridge classifier resulted in an accuracy of 31% thus reflecting overfitting of the decoding classifier).

In summary, we quantified the extent of generic topographical organization across all sublayers across the full Topoformer-BERT model, and honed in on selectivities of the topographic organization of the final layer. We analyzed eight different linguistic properties, finding strong effects for broad *semantic* dimensions of sentence content: animacy, concreteness, and visuomotor properties. It is important to note that despite the strongly significant selectivity, the mean activity patterns were highly similar across categories within each contrast (Appendix A.2.3, Figure 8): rather than indi-

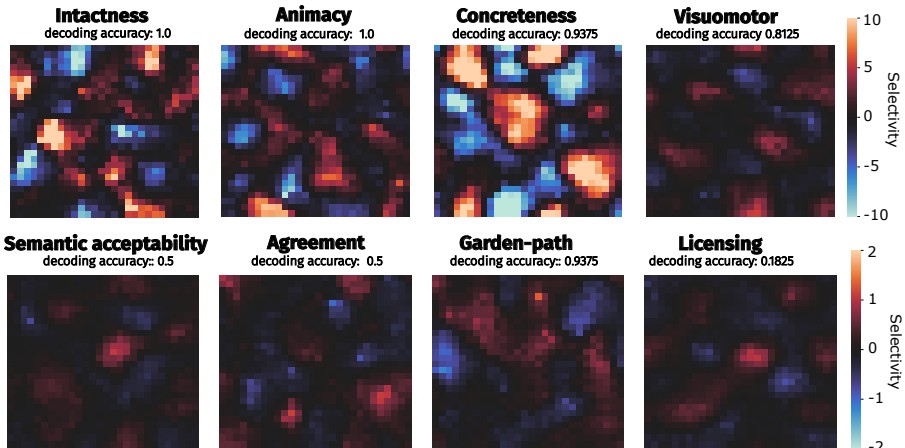

**Figure 4: Selectivity-based interpretation of topographic organization in Topoformer-BERT.**
Each panel shows the selectivity of Topoformer-BERT layer 15 (keys), for a given contrast. Each test suite contains two contrasting conditions each with a set of sentences; unit activities are computed as the mean over tokens for each sentence, and the conditions are contrasted with a $t$-test. We plot the selectivity significance value, as in Figure 2. The first row contains sentences with natural variability, whereas the bottom row contains results from constructed minimal pairs differing in only one word across conditions. To ensure visibility of effects regardless of size, we used different statistic ranges for plotting of each row: $s = 10$ for the top row, and $s = 2$ for the bottom row.

cating contrasting hot spots of activation for animate and inanimate content, the overall pattern of activation across units is similar across text from different conditions, and selectivity across conditions is indicative of small yet distinct deviations from the overall activation pattern. This property is not unique to the Topoformer (Appendix A.2.3, Figure 8E.), however, the Topoformer allows for a uniquely intuitive visualization of selectivity patterns in 2D, aiding interpretability. Future work should investigate the organization of finer-grained semantic dimensions, as well as more extensive tests for syntactic knowledge.

### 3.3 MODELING THE TOPOGRAPHIC ORGANIZATION OF THE HUMAN LANGUAGE NETWORK

To assess the topographic organization of language in the human brain, we recorded brain responses using event-related fMRI from N=5 participants (4 female, native English speakers) during a sentence reading task. Participants read 1,000 6-word, corpus-extracted sentences that were selected to maximize semantic and stylistic diversity (see Appendix A.8). Following standard preprocessing, we used a set of five anatomical language masks ("parcels") that denote brain regions within which most or all individuals in prior studies (Fedorenko et al., 2010; Lipkin et al., 2022) showed activity for an extensively validated language localizer contrast between reading of sentences and non-word strings (Fedorenko et al., 2010). For each participant, within these anatomical parcels, we then computed individual functionally-defined language regions by comparing responses to sentences and non-words, and taking all voxels with at least weak preferences for sentences over nonwords ($t > 1$). We then restricted our analyses to these voxels, henceforth the "language network". To determine that the language network exhibits spatial smoothness, we computed the generic topographic statistic $t_g$ (Equation 4) on unsmoothed brain responses within the language network of each participant, splitting the network into the five subregions demarcated by the anatomical parcels (see Appendix A.8). We found that the $t_g$ value for each cluster fell outside a null distribution computed using shuffled brain responses indicating significant decay in unit response correlations with distance – and therefore topographic organization – in the language-selective brain regions (which could not be explained by transformation into a template brain space, see Appendix A.9).

To determine whether the topography of the human language network is linguistically meaningful, and functionally related to that of Topoformer-BERT, we performed representational alignment using partial least squares singular value decomposition (PLS-SVD). Given z-scored brain responses $X$ and model embeddings $Y$, PLS-SVD finds joint low-dimensional embeddings $X_c$ and $Y_c$ by com-

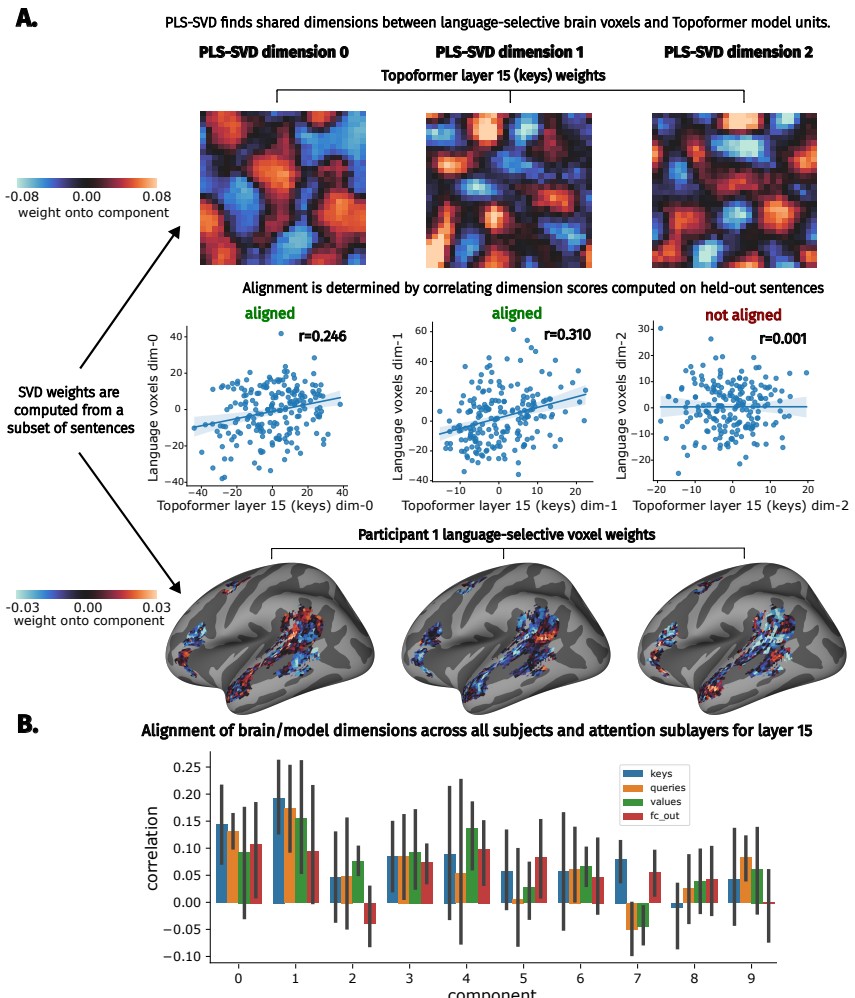

**Figure 5: Alignment of topographic representations in the human language network and Topoformer-BERT model. A.** Illustration of the PLS-SVD alignment approach for a single participant and model sublayer representation. **B.** Alignment quantified across all 10 components, and each sublayer of Topoformer-BERT layer 15. The alignment of components is computed as the correlation of respective cross-validated PLS-SVD component scores for brain and model representations. Error bars indicate 95% confidence intervals over 5 participants.

puting the SVD on the cross-covariance matrix: $X^T Y = U \Sigma V$. The component scores are then given as $X_c = X W_x$ and $Y_c = Y W_y$, where $X_c^{(i)}$ and $Y_c^{(i)}$ are the $i$-th aligned component scores. The critical test of alignment is whether new responses from each component are correlated between brain and model; a significant correlation would suggest that the component captures some shared linguistic variability. To test this, we computed PLS-SVD on 80% of the sentences, and correlated the component scores on the remaining 20% of sentences. To visualize the topography of these components, we note that the left singular vectors $U = W_x$ are the component weights from brain responses and the right singular vectors $V^T = W_y$ are the component weights from model embeddings. We can thus visualize individual brain and model components $W_x^{(i)}$ and $W_y^{(i)}$, respectively, reshaped into their native spatial format.

Figure 5A plots example alignments between the first three brain and model components, using the first participant and the Topoformer-BERT layer 15 (final layer, zero-indexed) keys representation (other sublayers shown in Appendix A.10.1). We see that the first two components are strongly aligned, as well as strongly topographically organized in both model and brain spaces. The third component is not aligned, despite being spatially organized in each representational space. Figure

5B repeats this analysis for all participants and sublayers, using layer 15 again. In general, the first two components were significantly aligned for each sublayer, whereas later components were less likely to be aligned. This result demonstrates that the low-dimensional variability can be aligned in the topographic representations of the human language network and Topoformer language model. The fact that we used functionally-defined language regions suggests that there is spatial functional organization even *within* this relatively functionally homogeneous brain network (e.g., Blank & Fedorenko (2020); Fedorenko et al. (2020)), rather than simply across different functional networks with heterogeneous response profiles.

To determine the specificity of this alignment, we performed an identical analysis using non-linguistic brain regions and an untrained Topoformer-BERT variant (Appendix A.10). Alignment, as well as voxel encoding model prediction (Appendix A.10.2), was significantly greater between the trained Topoformer-BERT and language network compared to non-linguistic control regions and an untrained model, highlighting the linguistic nature of the alignment. Notably, the *alignment* is not dependent on using a topographic rather than standard transformer, however, the Topoformer uniquely allows for a spatial visualization of the aligned component (Appendix A.10).

## 4 DISCUSSION

Here, we introduced the first topographically organized Transformer language models: "Topoformers". Across small and large models, we found that spatial querying and reweighting operations produced topographic organization in Topoformer models trained on NLP tasks. We quantified and analyzed the resulting topographic organization through hypothesized linguistic contrasts, as well as generically through PCA. Finally, we analyzed brain responses to a large number of sentences in the human language network, and demonstrated low-dimensional alignment of topographic variability with that found in the Topoformer-BERT model.

Introducing topography into language models may improve interpretability in NLP. We took some initial steps in interpreting the resulting topography in the Topoformer-BERT model based on previously hypothesized linguistic contrasts (e.g., Binder et al. (2005); Desai et al. (2020); Misra et al. (2023). These test suites could be extended depending on the question of interest. Importantly, the interpretability problem is far from solved. One issue is that of "polysemanticity", whereby units are involved in the representation of several distinct concepts (Bricken et al., 2023). Despite strong semantic selectivity, we found that Topoformer-BERT's activations were highly overlapping across categories, similar to non-topographic models. While our 2D visualizations aided interpretability of selectivity, the dimensions of organization in both the Topoformer and the human brain language network were not easily interpreted. Efforts to extract "monosemantic" features (Cunningham et al., 2023; Bricken et al., 2023) or to encourage disentangled representations (Higgins et al., 2021) may prove fruitful in yielding more interpretable topography in models, which in turn may help generate hypotheses regarding the human brain, where experiments are limited by cost and time.

Our work is limited in that it currently explores organization only within the functionally-defined language network. Previous works have also characterized semantic organization across the whole brain (Huth et al., 2012; 2016), and it would be intriguing to apply similar analyses to the ones we perform here to larger scale semantic organization in the brain. However, under the assumption that LLMs are good models of language, rather than human thought more broadly (Mahowald et al., 2023) – or grounded semantics for that matter (Mahon & Caramazza, 2008) – our choice is a principled first step towards characterizing the topographic organization of linguistic representations.

Additionally, limitations in brain data include that brain responses were acquired in a rapid, event-related fMRI design without any sentence repetitions, making the data susceptible to noise. Future work using data with multiple repetitions over specific linguistic contents should help to achieve more reliable and interpretable discovery of topographic variability. Additionally, other datasets targeted at examining specific hypotheses about linguistic organization will be useful.

This work marks the beginning of topographic modeling of language processing and there are several open questions to answer before a mature understanding of language topographic organization can be reached. Beyond language, as Transformers are a domain-general architecture, we expect topographic transformers to be a useful tool for modeling brain organization more broadly, particularly functional areas whose purported function can be performed well by transformers.

## CODE AVAILABILITY

Code to train Topoformer models is made available at https://github.com/TahaBinhuraib/topoformer. fMRI analysis code is available upon request.

## AUTHOR CONTRIBUTIONS

Conceptualization (model): TB, NMB. Software development (model): TB, NMB. Software development (fMRI analysis): GT, NMB. Interpretability tests: GT, TB. fMRI data collection: GT. Data analysis: TB, GT, NMB. Writing, original draft: TB, NMB. Writing, review and editing: TB, GT, NMB.

## ACKNOWLEDGMENTS

We thank Leila Wehbe, Mariya Toneva, David Plaut, Kendrick Kay, and the Harvard Vision Sciences Lab for feedback on earlier portions of this work. NMB acknowledges support from NSF grant #2123069. GT acknowledges support from the K. Lisa Yang Integrative Computational Neuroscience Center Graduate Fellowship. TB acknowledges Novus Technologies for the use of computing resources.

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

## A  APPENDIX

### A.1  INTUITIVE EXPLANATION OF SPATIAL QUERYING

The full self-attention equation with spatial querying (SQ) is given as follows:

$$\text{Attention}_{\text{SQ}}(Q, K, V) = \text{softmax}\left(\frac{QMK^T}{\sqrt{d_k}}\right) V \, W^O \tag{2}$$

The local pooling of spatial querying can be visualized with a simple example, assuming a model dimension of $d = 3$, and 2 tokens.

$$QMK^T = \begin{pmatrix} Q_{1,1} & Q_{1,2} & Q_{1,3} \\ Q_{2,1} & Q_{2,2} & Q_{2,3} \end{pmatrix} \begin{pmatrix} 1 & 0 & 1 \\ 1 & 1 & 0 \\ 0 & 1 & 1 \end{pmatrix} \begin{pmatrix} K_{1,1} & K_{2,1} \\ K_{1,2} & K_{2,2} \\ K_{1,3} & K_{2,3} \end{pmatrix} \tag{3}$$

$$= \begin{pmatrix} Q_{1,1} + Q_{1,2} & Q_{1,2} + Q_{1,3} & Q_{1,1} + Q_{1,3} \\ Q_{2,1} + Q_{2,2} & Q_{2,2} + Q_{2,3} & Q_{2,1} + Q_{2,3} \end{pmatrix} \begin{pmatrix} K_{1,1} & K_{2,1} \\ K_{1,2} & K_{2,2} \\ K_{1,3} & K_{2,3} \end{pmatrix}$$

We can see that, instead of the rows of the matrix multiplication containing individual queries, they now contain summed local pools of queries. Spatial querying can easily be generalized to encompass learned pooling, by converting $M$ into a learnable locally connected matrix. We leave this to future work.

### A.2  RECEPTIVE FIELD (RF) SIZE ANALYSIS

#### A.2.1  EFFECT ON PERFORMANCE

In this section, we discuss our methodology for choosing a receptive field (RF) size. First, we train the 1-layer (**Topoformer-SQR**) across a sweep of different RF sizes for both spatial querying ($r_{SQ}$) and spatial reweighting ($r_{SR}$). As seen in Figure 6, we can see a clear inverse relationship between the model's performance on the IMDB sentiment test set and the $r_{SQ}$ ($R^2 = 0.8612$), in line with the idea that larger spatial querying pools reduce the amount of information due to the local averaging of queries within the pool. However, we found that the size of spatial reweighting $r_{SR}$ has no measurable effect on model performance $r_{SR}$ ($R^2 = 0.0137$). This relationship deviates from the spatial querying result because there is no averaging performed in spatial reweighting. Future work could explore a learned/weighted version of spatial querying that would be expected to show less of a decrement in performance with increasing RF size. Importantly, substantial topographic organization can be seen with a very small SQ value (see Figure 7), such that performance decrements are minimal.

**A** Spatial querying

**B** Spatial reweighting

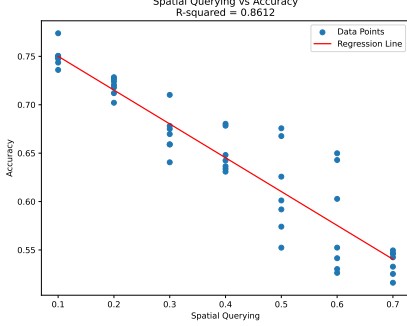
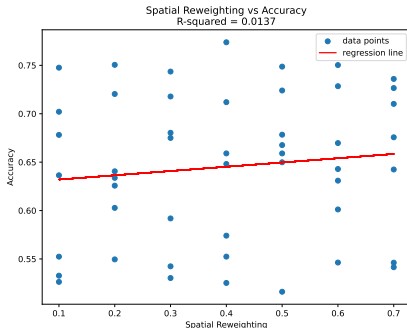

**Figure 6:** RF size effect on accuracy for the IMDB sentiment test set.

### A.2.2 QUALITATIVE EVALUATION

Using the Topoformer-SQR models trained across a range of RF values from the previous section (A.2.1), we visualized the topographical organization across the different RF values. To do so, we plotted a matrix grid containing the weights of the second PC across all RF values (given that this PC seemingly aligned with the sentiment selectivity, see Figure 2). For brevity, we visualized the the keys and fc_out sublayer representations using the same procedure mentioned in Figure 2, so as to determine the effects of RF size on both spatial querying and reweighting. As seen in Figure 7, having smaller $r_{SQ}$ values results in much stronger topographic organization in the keys, in line with the better performance; larger values of $r_{SQ}$ were associated with poor performance and minimal organization, suggesting that the model struggled to learn representations in the presence of large averaging pools. However, also similar to the performance results, the spatial reweighting RF $r_{SR}$ had minimal effect on the topography, with topographic organization developing in the fc_out sublayer across a large range of RF sizes.

**A** Keys PC2

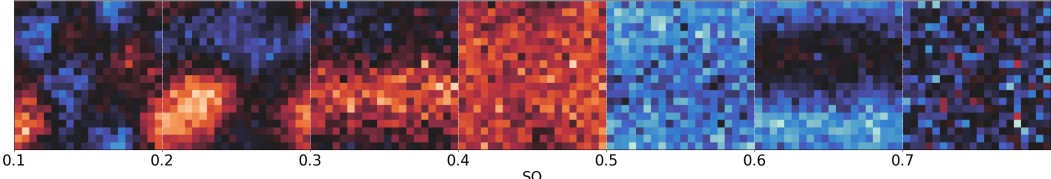

**B** fc_out PC2

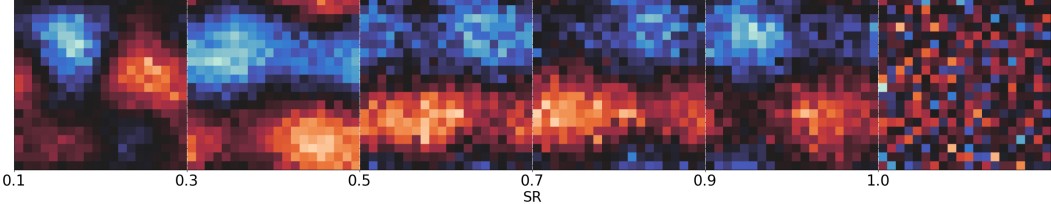

**Figure 7:** Effect of receptive field size on topographies. Panel **A** illustrates the impact of the spatial querying RF width $r_{SQ}$ on the topography of keys, while panel **B** visualizes the effect of spatial reweighting RF width $r_{SR}$ on the topography of fc_out.

### A.2.3 MEAN UNIT ACTIVATIONS

In this section, we extract mean unit activations over tokens for each sentence in each condition from two different test suites, yielding a mean activation profile for each of the four conditions separately. As illustrated in Figure 8, these mean activation profiles exhibit a high degree of correlation. To explore whether this phenomenon is unique to Topoformer-BERT, we conduct a comparative analysis of keys sublayer representations for animate and inanimate sentences across three distinct Transformer models (non-topographic BERT control model, Topoformer-BERT, and an off-the-shelf sentenceBERT model). The results, depicted in Figure 8, reveal a high correlation among models, including an off-the-shelf all-mini-LM-L6-v2 (Reimers & Gurevych, 2019), demonstrating consistent responses across different categories.

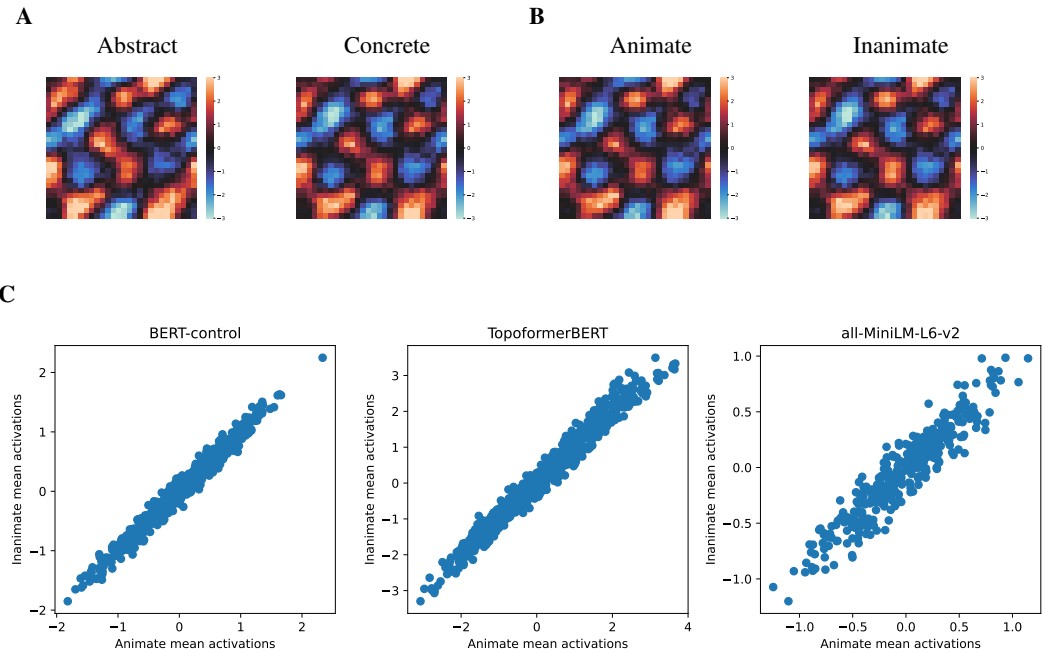

**Figure 8:** Mean unit activations in Topoformer-BERT layer 15 (keys) across two different conditions from each of two test suites (abstractness (**A.**) and animacy (**B.**)). Mean activations are highly correlated across all categories. Panel **C** Compares the keys sublayer activations for sentences about animate and inanimate content, across three different models in the final layer of Topoformer-BERT, BERT-control, and an off-the-shelf model all-miniLM-L6-v2 (Reimers & Gurevych, 2019) activations. Each model shows highly correlated responses.

### A.3 FEASIBILITY MODEL METHODOLOGY (TOPOFORMER-SQ AND TOPOFORMER-SQR)

### A.3.1 TRAINING PARADIGM

We trained a 1-layer encoder-only Transformer on the IMDB sentiment analysis dataset (Maas et al., 2011), which classifies movie reviews as having a positive or negative overall sentiment. We utilized the average of all tokens as the input to a binary classifier, as is standard practice, and optimized a cross-entropy loss. We trained for 20 epochs, which was sufficient to reach convergence. We used an Adam optimizer (Kingma & Ba, 2017) with a learning rate of 0.001. A batch size of 128 was used for the SQ model, while a batch size of 256 was used for the SQR model (SQR was less stable and required the larger batch size; similar results were seen across batch sizes for SQ).

### A.3.2 SELECTIVITY ANALYSES

After completing the training process, we conducted a selectivity analysis which is ubiquitously used in neuroscience. We probed the Q, K, and V layers to gain insights into their topographic organization and how the model attends to different aspects of the input data. By analyzing the responses of each domain (i.e., sentences with positive and negative sentiment) versus the others using a two-tailed t-test, we obtained the test statistic $t$, the significance value $p$ of the test, and the sign of the test statistic $s = \text{sign}(t)$. We then computed the selectivity as $-s\log_{10}(p)$, which provided a quantification of our model's selectivity for each domain.

### A.4 GENERAL ANALYSIS METHODOLOGY

### A.4.1 PRINCIPAL COMPONENT ANALYSIS

In addition to selectivity analyses, for all models, as well as brains, we aimed to uncover generic patterns of activation in the layers of our model without any specific hypothesis in mind. To this end, we performed principal component analysis (PCA) on the activations of each layer individually. By analyzing the PCs, we were able to identify the dominant modes of variation in the activation data, and gain deeper insights into the structure of the activations. To visualize these patterns of activation, we reshaped the weights of the first two PCs to the same size as our cortical map. This allowed us to compare the patterns of activation across different layers and gain a deeper understanding of the topographic organization of our model.

### A.4.2 QUANTIFICATION OF TOPOGRAPHY

We compute the generic topographic statistic $t_g$ as a measure of the general distance dependence of pairwise response correlations, based on the notion that local correlation is a hallmark of topographic organization (Kiani et al., 2007; Lee et al., 2020b). Given the Pearson correlation matrix $R_{i,j} = r_p(\boldsymbol{a}_i, \boldsymbol{a}_j)$ — where $\boldsymbol{a}_i$ gives the activity vector of unit $i$ over sentences, $r_p(\boldsymbol{x}, \boldsymbol{y})$ gives the Pearson correlation of $\boldsymbol{x}$ and $\boldsymbol{y}$, and $r_s(\boldsymbol{x}, \boldsymbol{y})$ gives the Spearman rank correlation of $\boldsymbol{x}$ and $\boldsymbol{y}$ — and the pairwise Euclidean distances $\mathcal{D}_{i,j}$ computed in volumetric space, along with a maximum distance over which to compute the statistic $d$, we compute the generic statistic as follows:

$$t_{g,d} = r_s(-R_{i,j}, \mathcal{D}_{i,j}) \ \forall \ i, j : D_{i,j} < d \tag{4}$$

Because topographic organization can exist at multiple scales, with long-range correlation violating the locally distance-decaying correlation, we may wish to compute $t_{g,d}$ at a range of maximum distances, as in Figure 3B. Thus, for those analyses, we computed a vector $\boldsymbol{t}_g = \{t_{g,d_0}, ..., t_{g,d_n}\}$ over a linearly spaced range maximum distance values $\boldsymbol{d}$. We summarized this vector by taking the mean $\bar{t}_g = \frac{1}{n}\sum_i \boldsymbol{t_g^i}$ (Figure 3A), and plotting the vector against $\boldsymbol{d}$ (Figure 3B). When no maximum distance is used, we refer to the statistic as simply $t_g$, as used in the brain analyses.

### A.5 LACK OF TOPOGRAPHIC ORGANIZATION IN THE CONTROL MODEL

We trained a single-head BERT model using the same dataset and training procedure as employed for the Topoformer-BERT model. Notably, the control model was trained with the intentional exclusion of spatial querying and reweighing operations. All other aspects of the model, including architecture and training parameters, remained consistent with those of the Topoformer-BERT. Subsequently, we followed the outlined procedure depicted in Figure 2 to extract activations corresponding to sublayer representations from the control model.

As expected, the outputs from this non-topographic BERT control model exhibited a lack of discernible topographic organization. This deficiency arises due to the absence of spatial querying or reweighting mechanisms within the model architecture.

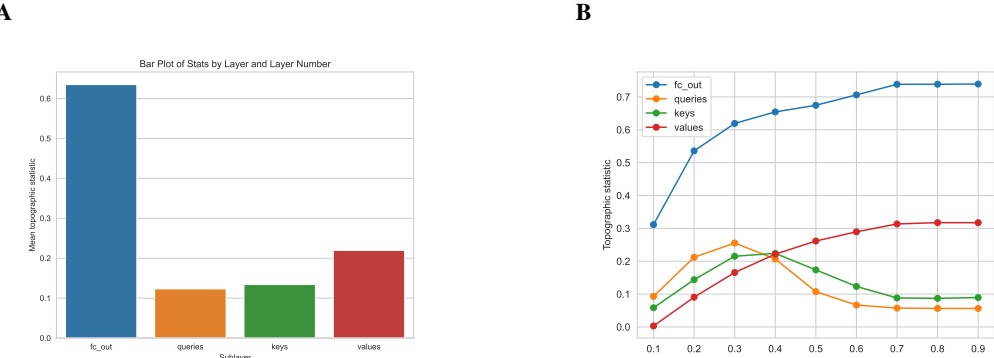

**Figure 9: Topographic organization across sublayers of Topoformer-SQR.**
**A.** We quantified the extent of topography in Topoformer-BERT via the generic topography statistic $\bar{t}_g$ (Equation 4) **B.** The topography statistic $t_{g,d}$ for all sublayers computed at each of a range of scales.

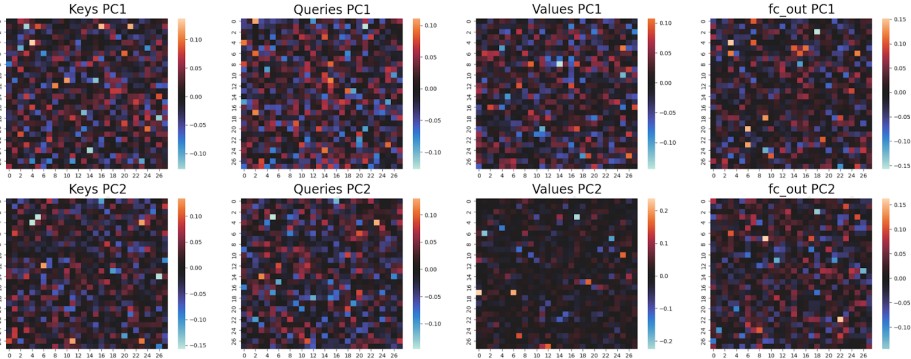

**Figure 10:** Lack of topographic organization in the control model. Here, we can see that none of the (keys, queries, values, and fc_out) were topographically organized.

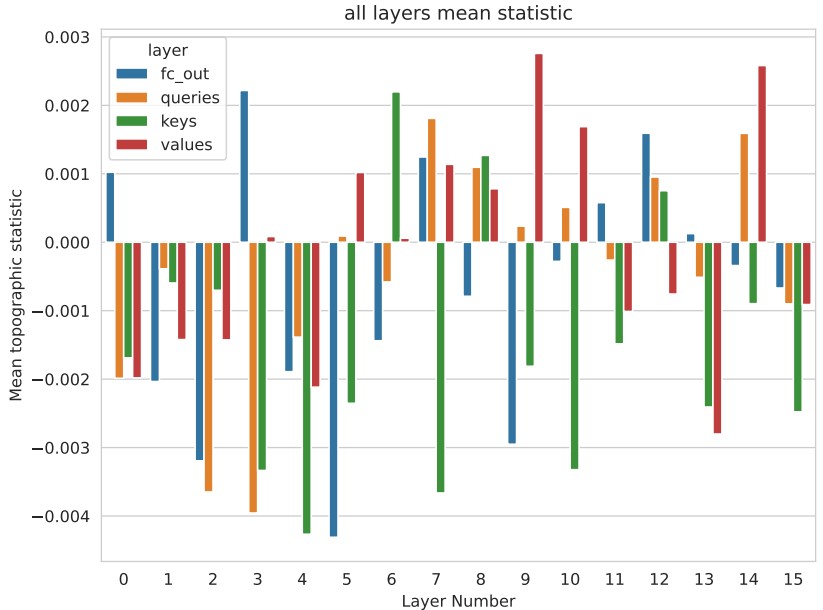

**Figure 11: Lack of topographic organization across all layers of Topoformer-BERT.**
We quantified the extent of topography in Topoformer-BERT via the generic topography statistic $\bar{t}_g$ (Equation 4) which intuitively relates the degree of correlation with the distance between pairs of units averaged over a range of scales (nine maximum distance values).

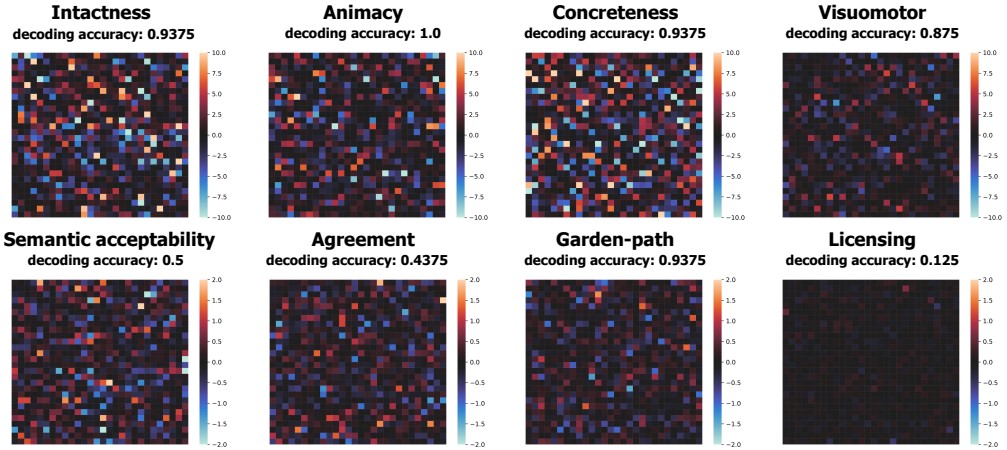

**Figure 12: Selectivity-based interpretation of non-topographic organization in the control model.** The similar decoding accuracies to the Topoformer-BERT suggest that the information is contained, but organized differently across test suites.(e.g., Licensing, Agreement in Figure 12) were also reflected in poorer decoding over the patterns of key activations (chance-level 50%; in the case of Licensing, the use of a Ridge Classifier resulted in an accuracy of 43%, while in Agreement an accuracy of 50% was observed, thus reflecting overfitting).

## A.6 LARGE-SCALE MODEL METHODOLOGY (TOPOFORMER-BERT)

### A.6.1 TRAINING METHODOLOGY

We used a batch size of 4096, and Adam optimizer (Kingma & Ba, 2017) with weight decay of 0.01, $\epsilon = 10^{-12}$, $\beta_1 = 0.9$ $\beta_2 = 0.98$, gradient clipping at a clip value of 0.5.

### A.6.2 GENERIC TOPOGRAPHY ANALYSES

To analyze generic topography (Figure 3), we sampled random sentences from the same Bookcorpus-Wikipedia dataset that was used in training. The activations of 300 random sentences from the Bookcorpus-Wikipedia dataset were extracted and used to analyze generic topography.

### A.6.3 SELECTIVITY ANALYSES

Since the large-scale models were not trained on a categorization task, we had to devise a particular hypothesis of linguistic relevance for the large-scale Topoformers. To accomplish this, we investigated eight test suites that each target different properties of linguistic input (see Section 3.2 and Appendix A.7). We note that, in contrast to the generic topography analyses, for these selectivity analyses, independent sets of sentences were used (not from the Bookcorpus-Wikipedia dataset), still demonstrating topographical organization (Figure 4).

To obtain the decoding accuracy scores showcased in Figure 4, we employed Principal Component Analysis (PCA) as a dimensionality reduction technique on the activations of the keys sublayer. We then utilized 50 principal components to train a logistic regression classifier for decoding the two different conditions within a test suite. For instance, in the case of **Animacy**, our objective was to predict whether the activations originated from an animate or inanimate sentence. The dataset underwent a fixed 80-20 split for training and testing, respectively.

## A.7 TEST SUITES

We evaluated a set of eight test suites targeting different linguistic properties. An overview of these suites along with examples can be seen in Table 2.

| Test Suite | Category | Example |
|---|---|---|
| Intactness | Intact | She scored 2 goals in the soccer game. |
| | Scrambled | Soccer scored game. the She in 2 goals. |
| Animacy | Animate | The gnu galloped across the savanna, majestic and swift. |
| | Inanimate | The oven's warm glow promised delicious, freshly baked bread. |
| Concreteness | Concrete | She peeled the banana slowly, savoring its sweet, ripe aroma. |
| | Abstract | Her motive for volunteering was purely altruistic and kind. |
| Visuomotor | Visual | To solve problems, I often visualize them in my mind. |
| | Motor | His grip on the rope tightened as he climbed higher. |
| Semantic Acceptability | Acceptable | A sunflower has yellow petals. |
| | Unacceptable | A peanut has yellow petals. |
| Agreement | Matched | The authors that hurt the senator are good. |
| | Mismatched | The authors that hurt the senator is good. |
| Licensing | Matched | The authors that liked the senator hurt themselves. |
| | Mismatched | The authors that liked the senator hurt himself. |
| Garden-Path | Ambiguous | As the criminal shot the woman with her young daughters yelled at the top of her lungs. |
| | Unambiguous | As the criminal fled the woman with her young daughters yelled at the top of her lungs. |

**Table 2:** Overview of test suites with sentence examples. Each test suite had 38 sentences in each category, for a total of 76 sentences in each suite.

All eight test suites consisted of 76 sentences each, with 38 sentences in each category.

The first suite, **Intactness** consisted of intact sentences versus their scrambled counterparts, thereby degrading both linguistic meaning (syntax) and meaning (semantics). Capitalization and final sentence punctuation was retained in the scrambled sentences. Suites 2 through 4 evaluated three different dimensions of *meaning* that have been extensively investigated in prior work, as specified next. Suite 2, **Animacy**, consisted of sentences with animate vs. inanimate meanings (Naselaris et al., 2009; Connolly et al., 2012; Konkle & Caramazza, 2013). We sampled 76 animate/inanimate word categories from (Konkle & Caramazza, 2013). Specifically, 38 animate words were randomly sampled from the "Big-Animate" and "Small-Animate" categories (from 120 words in total), and 38 inanimate words were randomly sampled from the "Big-Inanimate", and "Small-Inanimate" categories (from 120 words in total). ChatGPT (version 4) was prompted to generate sentences about each of these words, approximately 10 words long. Suite 3, **Concreteness**, consisted of sentences with concrete vs. abstract meanings (Binder et al., 2005; Fiebach & Friederici, 2004). We randomly sampled 38 concrete and 38 abstract word categories from (Binder et al., 2005) (from 50 words in total in each category), and similarly prompted ChatGPT to generate approximately 10-word-long sentences about each of these words. Suite 4, **Visuomotor**, consisted of sentences with visual vs. motor meanings (Desai et al., 2010; Lynott et al., 2020). We took all available visual and motor verbs from (Desai et al., 2010) (23 in each category). For the remaining 15 words for each category (to obtain 38 sentences in each category to unify the number of sentences across suites), we sampled words from the Lancaster Sensorimotor Norms (Lynott et al., 2020). Specifically, for the visual category, we selected the 15 top-rated "Visual" words. For the motor category, we averaged across the "Foot leg", "Hand arm", "Head", "Mouth", and "Torso" ratings and selected the 15-top rated words excluding inappropriate words and different forms of the same word stem. We similarly prompted ChatGPT to generate approximately 10-word-long sentences about each of these words.

For all the three semantic test suites (2, 3, 4) we tested that the surprisal of the sentences in each category (within a suite) were not significantly different from each other to avoid a confound of overall sentence surprisal (evaluated by two-sided, unpaired t-tests; all three $p > 0.11$). Sentence surprisal was estimated using GPT2 (Radford et al. (2018) (from HuggingFace, Wolf et al. (2020) via the "surprisal" package: `https://github.com/aalok-sathe/surprisal`) as the average of the surprisal values for each token in the sentence.

Suite 5, **Semantic acceptability** consisted of minimal pair sentences (Conceptual Minimal Pair Sentences Base; Misra et al. 2023) with 38 acceptable sentences and 38 unacceptable sentences. The remaining three suites (6, 7, 8) evaluated three different dimensions of *form* using minimal pair test suites from SyntaxGym (Gauthier et al., 2020; Hu et al., 2020): Suite 6 consisted of matched/mismatched **Agreement** sentences (Subject-Verb Number Agreement; `https://syntaxgym.org/test_suite/items?test_suite=261`), suite 7 consisted of matched/mismatched **Licensing** sentences (Reflexive Number Agreement; `https://syntaxgym.org/test_suite/items?test_suite=260`), and suite 8 consisted of **Garden-Path** ambiguous sentences (Verb Transitivity; `https://syntaxgym.org/test_suite/items?test_suite=270`).

## A.8 HUMAN BRAIN DATA

### A.8.1 PARTICIPANTS AND ACQUISITION

We recorded brain responses using fMRI from N=5 participants during a sentence reading task (Tuckute et al., 2024). The participants were neurotypical native speakers of English (4 female), aged 21 to 30 (mean 25; std 3.5), all right-handed. Participants read 1,000 6-word, corpus-extracted sentences that were selected to maximize semantic and stylistic diversity. Participants completed two scanning sessions where each session consisted of 10 runs of the sentence reading experiment (sentences presented on the screen one at a time for 2s with an inter-stimulus interval of 4s, 50 sentences per run) along with additional tasks. Participants were exposed to the same set of 1,000 sentences (no repetitions), but in fully randomized order. Structural and functional data were collected on the whole-body, 3 Tesla, Siemens Prisma scanner with a 32-channel head coil. T1-weighted, Magnetization Prepared RApid Gradient Echo (MP-RAGE) structural images were collected in 176 sagittal slices with 1 mm isotropic voxels (TR = 2,530 ms, TE = 3.48 ms, TI = 1100 ms, flip = 8 degrees). Functional, blood oxygenation level dependent (BOLD) were acquired using an SMS EPI sequence (with a 90 degree flip angle and using a slice acceleration factor of 2), with the following acquisition parameters: fifty-two 2 mm thick near-axial slices acquired in the interleaved order (with 10% distance factor) 2 mm × 2 mm in-plane resolution, FoV in the phase encoding (A ≪ P) direction

208 mm and matrix size 104 × 104, TR = 2,000 ms and TE = 30 ms, and partial Fourier of 7/8. All participants gave informed written consent in accordance with the requirements of an institutional review board.

### A.8.2  DATA PREPROCESSING AND FIRST-LEVEL MODELING

fMRI data were preprocessed using SPM12 (release 7487), and custom CONN/MATLAB scripts. Each participant's functional and structural data were converted from DICOM to NIfTI format. All functional scans were coregistered and resampled using B-spline interpolation to the first scan of the first session. Potential outlier scans were identified from the resulting participant-motion estimates as well as from BOLD signal indicators using default thresholds in CONN preprocessing pipeline (5 standard deviations above the mean in global BOLD signal change, or framewise displacement values above 0.9 mm; (Nieto-Castañón, 2020)). Functional and structural data were independently normalized into a common space (the Montreal Neurological Institute [MNI] template; IXI549Space) using SPM12 unified segmentation and normalization procedure (Ashburner & Friston, 2005) with a reference functional image computed as the mean functional data after realignment across all timepoints omitting outlier scans. The output data were resampled to a common bounding box between MNI-space coordinates (-90, -126, -72) and (90, 90, 108), using 2 mm isotropic voxels and 4th order spline interpolation for the functional data, and 1 mm isotropic voxels and trilinear interpolation for the structural data. Last, the functional data were not spatially smoothed (to ensure that are claims about topographic organization could not be explained by smoothing). A General Linear Model (GLM) was used to estimate the beta weights that represent the blood oxygenation level dependent (BOLD) response amplitude evoked by each individual sentence trial using GLMsingle (Prince et al., 2022) (fixation was modeled implicitly, such that all timepoints that did not correspond to one of the conditions (sentences) were assumed to correspond to a fixation period). Within the GLMsingle framework, the HRF which provided the best fit to the data was identified for each voxel (based on the amount of variance explained). Data were modeled using 5 noise regressors and a ridge regression fraction of 0.05. The 'sessionindicator' option in GLMsingle was used to specify how different input runs were grouped into sessions. By default, GLMsingle returns beta weights in units of percent signal change by dividing by the mean signal intensity observed at each voxel and multiplying by 100. Hence, the beta weight for each voxel can be interpreted as a change in BOLD signal for a given sentence trial relative to the fixation baseline. After first-level modeling, the voxels within a set of 5 masks ("parcels") were extracted. These parcels were derived from n=220 independent participants using a Group-Constrained Subject-Specific (GSS; (Julian et al., 2012) based on an extensively validated language localizer contrast between reading of sentences and non-word strings (Fedorenko et al., 2010; Mahowald & Fedorenko, 2016; Lipkin et al., 2022). These parcels delineate the expected gross locations of language-selective brain regions but are sufficiently large to encompass individual variability. The parcels are in the left hemisphere, three frontal parcels (inferior frontal gyrus [IFG], its orbital portion [IFGorb], and middle frontal gyrus [MFG]) and two temporal ones (anterior temporal [AntTemp], posterior temporal [PostTemp]). The mean number of voxels in these five parcels were (they differ slightly across participants because of lack of spatial coverage in the functional acquisition sequence): IFG=743 (SD=0); IFGorb=364 (SD=13.4); MFG=462 (SD=0); AntTemp=1623 (SD=4.1); PostTemp=2948 (SD=0). The parcels are available for download at `https://evlab.mit.edu/funcloc/`. As a control brain region, we extracted voxels within a set of motor- and supplementary motor areas in the left hemisphere. Specifically, we used the Glasser parcellation (Glasser et al., 2016) to extract responses within 5 motor parcels: 1, 2, 3a, 3b, and 4. Moreover, we identified a set of supplementary regions using the grouping category "Paracentral lobular and mid-cingulate cortex" in (Tait et al., 2021) which consisted of 8 additional parcels: 24dd, 24dv, 6mp, 6ma, SCEF, 5m, 5L, and 5mv, yielding a total of 13 Glasser parcels of interest. The mean number of voxels in these 13 parcels were 3753 (SD=363.2).

## A.9  QUANTIFYING BRAIN TOPOGRAPHY

Here, we quantified the spatial smoothness of brain representations using the generic topographic statistic $t_g$, in each language ROI and a control ROI (see A.8 for details on ROI definition). Due to differences in sizes across ROIs, we use the version computed without a maximum distance, over the full range of voxel pairs in each ROI. Statistics are shown in Figure 13A. To determine the significance of these statistic values, which were low in some cases, a null permutation analysis was performed by shuffling the voxel responses of each region 100 times with respect to their positions, and computing $t_g$. This allowed us to construct a null distribution against which to compare real $t_g$ values. As can be seen in Figure 13B, the variability of this null distribution is very small. All of the brain data fell outside the 95th percentile, indicating that each brain region — including the control region — exhibited significant spatial smoothness in responses to natural sentences.

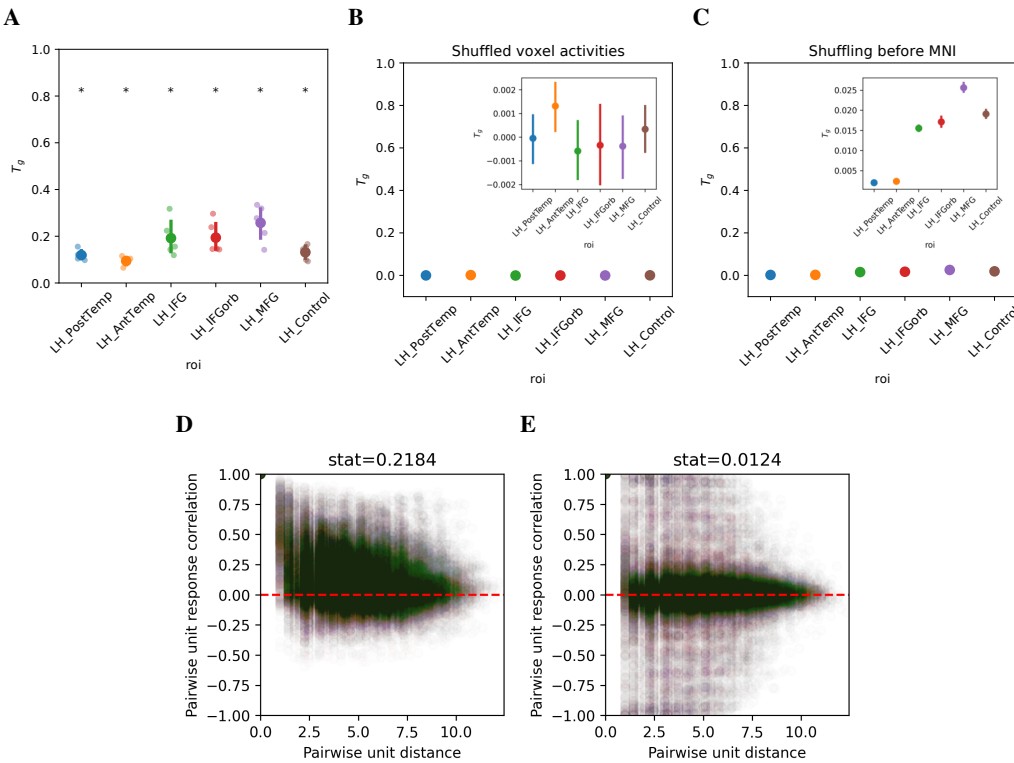

**Figure 13:** Generic topographic statistic of unsmoothed brain responses. **A.** Generic topographic statistic ($t_g$) for each ROI of the language network, and a control ROI. Error bars plot 95% confidence intervals. Stars indicate significance compared to both null distributions shown in **B.** and **C.**, using $\alpha = 0.05$. **B. and C.** Null distributions of $t_g$ computed for each ROI using shuffled voxel locations with 100 permutations per participant and ROI. Error bars plot 95% confidence intervals. In **B.**, voxel locations were shuffled within ROIs after all pre-processing. In **C.**, voxel locations were shuffled across the cortex after pre-processing but before transformation to MNI space. **D.** Example local correlation plot for one subject, using LH_MFG. **E.** Example pre-MNI shuffled local correlation plot for the same subject and ROI as in **D.**

As the transformation from native to group MNI space is expected to introduce some spatial blurring, we performed an additional control analysis to determine whether this transformation could account for the smoothness we observed within the language network. Here, before normalizing the data to MNI space, we first shuffled the whole-brain cortical responses (defined as voxels that were estimated as grey or white matter with probability $p > 0.5$). We then projected to MNI as before, using a trilinear alignment, and analyzed the voxels as in the main analyses.

As seen in Figure 13C and E., we found that the MNI transformation introduced a degree of very local correlation, but also a strong degree of very local anti-correlation, such that the $t_g$ statistic of shuffled data was very close to 0 for permuted data. To visualize the smoothness introduced by the MNI transformation, we performed PCA on the shuffled responses within the language network, and visualized the weights. We show an example subject in Figure 14, but the results were highly similar across all participants. Thus, we conclude that any smoothness introduced by the MNI transformation is insufficient to account for the smoothness we observe in the language network. While other sources of non-functional smoothness are possible – including intrinsic EPI blur, biological motion (e.g., pulse), and head motion – accounting for them is nontrivial and beyond the scope of this work.

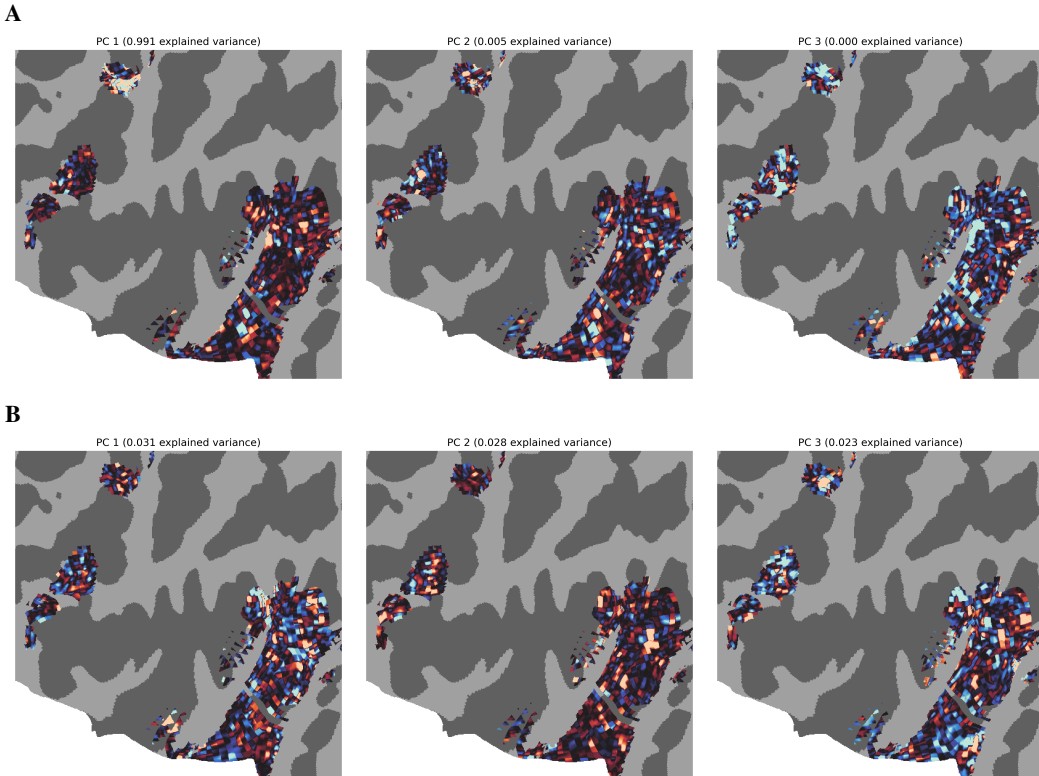

**Figure 14:** PCA weights for shuffled brain responses prior to MNI transformation. The MNI transformation introduces a small but negligible degree of spatial smoothness. **A.** and **B.** show 2 example subjects.

## A.10 BRAIN-MODEL CONTROL ANALYSES

In the main text, we performed a PLS-SVD analysis to assess the alignment of human language network and model components. Here, we repeated this analysis considering two important controls: i) motor-related control brain regions, and ii) untrained Topoformer-BERT. First, we confirmed that the alignment with Topoformer-BERT (Figure 15A) is stronger in the language network than in a control brain network (motor-related regions, see Appendix A.8 for details; Figure 15B). Particularly, whereas significant alignment was seen for the first two components in the language network, this was not true for most sublayers when comparing to the control brain network (exception: weak significant alignment of values with component 0, and keys with component 1) with substantially reduced alignment overall. Second, we analyzed an untrained version of the Topoformer-BERT model. We found that this untrained model demonstrated some alignment with the first (0th) component, however this was highly variable across participants and not statistically significant (Figure 15C). When comparing to the control brain network, the untrained model showed very little alignment (Figure 15D). Thus, the alignment was particularly strong between the trained Topoformer-BERT model and language network.

An additional question is whether the alignment should be greater between Topoformer-BERT vs. a non-topographic BERT and the language network. We hypothesized that their would be little difference in quantative component alignment, but that the components would only show a meaningful spatial arrangement in the Topoformer variant. Indeed, this is what we found, as shown in Figure 16. This highlights that the PLSSVD approach knows nothing about spatial layout per se in its component extraction, rather, the spatial organization is simply a useful and interesting feature for visualizing the components in 2D.

**A** Trained Topoformer-BERT, language network

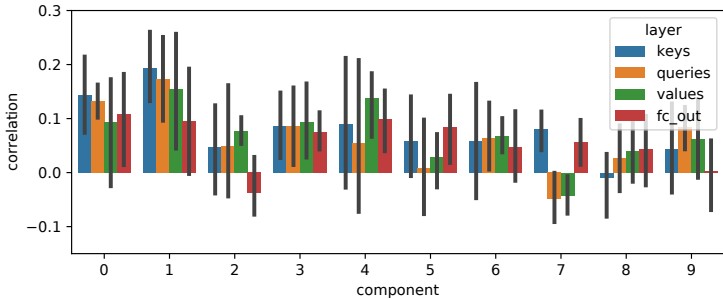

**B** Trained Topoformer-BERT, control brain regions

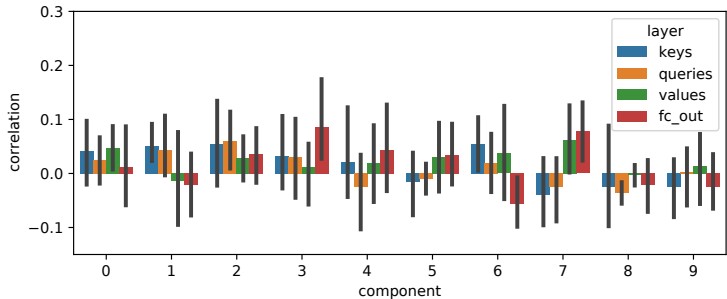

**C** Untrained Topoformer-BERT, language network

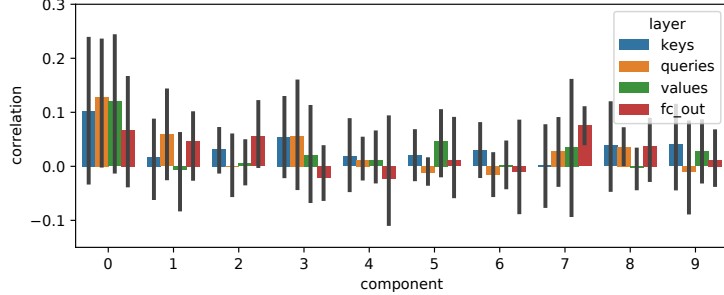

**D** Untrained Topoformer-BERT, control brain regions

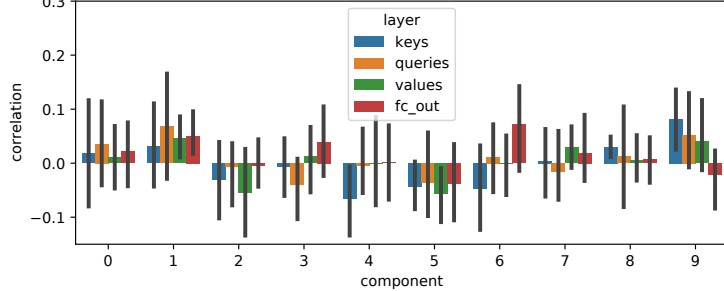

**Figure 15:** PLS-SVD alignment results across two control analyses. We follow the same analysis approach used in Figure 5, for each combination of trained/untrained Topoformer, and language/control brain regions. Error bars show 95% CI over all participants' voxels.

**A** Trained non-topographic BERT, language regions

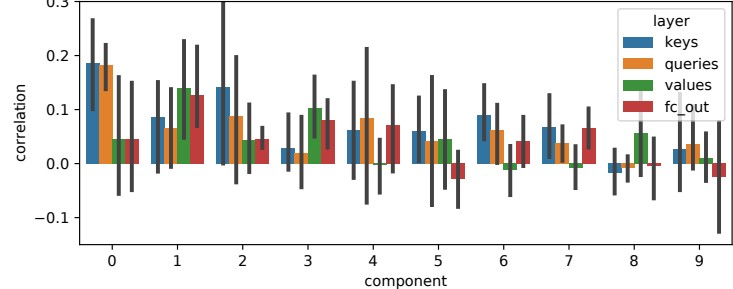

**B** Comparing Topoformer-BERT and control-BERT for the first 3 components

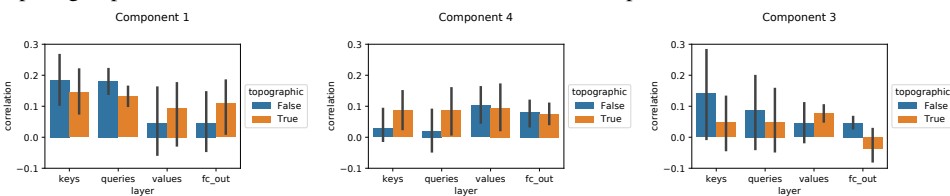

**C** PLS-SVD component weights for control-BERT

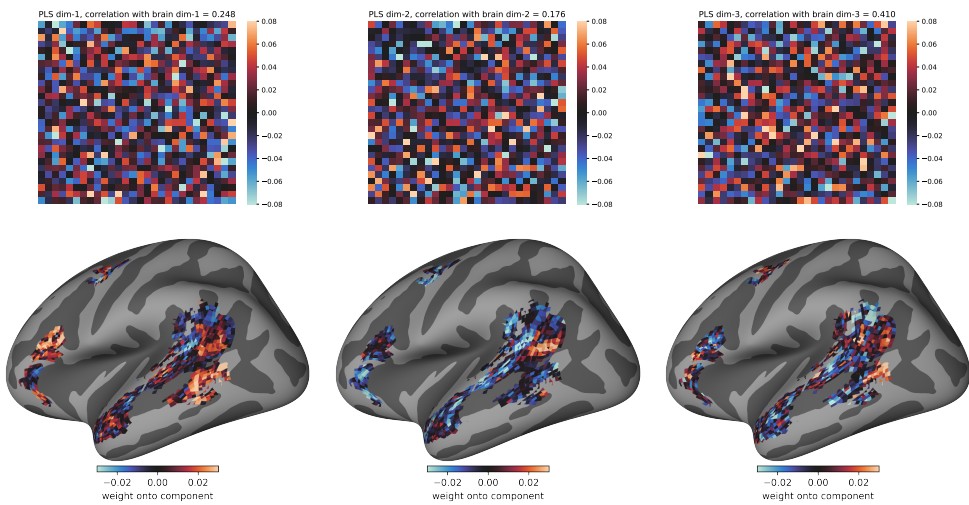

**Figure 16:** PLS-SVD alignment results using a non-topographic control BERT model.

### A.10.1   ADDITIONAL PLSSVD VISUALIZATIONS FOR TOPOFORMER-BERT

**A** PLS-SVD component weights for Topoformer-BERT fc_out



**B** PLS-SVD component weights for Topoformer-BERT queries

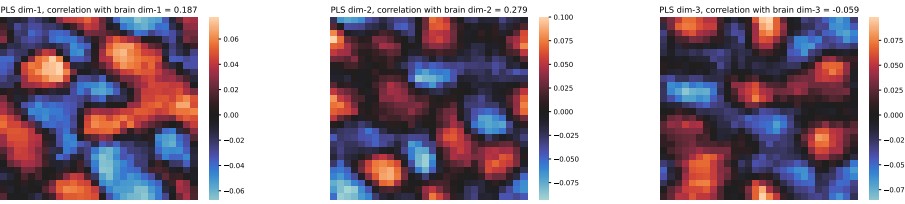

**C** PLS-SVD component weights for Topoformer-BERT values



**Figure 17:** PLS-SVD alignment example results for the other sublayers of Topoformer BERT layer 15. Analyses were performed as in Figure 5A, using the same subject. Results were similar across other subjects.

### A.10.2   ENCODING MODEL ANALYSES

Since our PLS-SVD alignment analysis is less common for comparing representations in the neuroscience literature than other analyses, we opted to perform an additional encoding model analysis for greater comparability to prior work.

The encoding model analysis approach is used to predict a given voxel's response based on a representational space. We employed ridge regression, or L2-regularized least squares regression. Formally, let us consider the model embedding space $X \in \mathbb{R}^{n \times d}$, where each of $n$ rows is a $d$-dimensional vector consisting of the model's representation of a given sentence. A given voxel's responses are given as $y \in \mathbb{R}^n$. For each voxel, we can formulate our regularized regression problem as finding a vector $\hat{w} \in \mathbb{R}^d$ such that

$$\hat{w} = \arg\min_{w} \|Xw - y\|_2^2 + \lambda\|w\|_2^2, \tag{5}$$

where $\|\cdot\|_2$ is the Euclidean norm. The $\lambda$ multiplier is a hyperparameter that specifies the relative weight of the regularization term in the loss. This value is chosen using leave-one-sentence-out cross-validation using 80% of the total data, as summarized below

$$\hat{\lambda} = \arg\min_{\lambda} \frac{1}{n} \sum_{k=1}^{.8*n} \left[ \|X^{(k)}\hat{w} - y^{(k)}\|_2^2 + \lambda\|\hat{w}\|_2^2 \right], \tag{6}$$

where $X^{(k)}$ and $y^{(k)}$ are held out cross-validation data at the $k$-th fold. This cross-validation is performed efficiently using the scikit-learn function RidgeCV (Pedregosa et al., 2011). The remaining 20% of the data is used to test the generalization of the encoding model, by predicting held-out voxel responses. We report the correlation of held-out voxel responses with predictions, and take the average over all voxels from all participants.

We perform this encoding model analysis for both trained and untrained Topoformer-BERT models, for voxels in both language and motor-related, non-linguistic control brain networks. The results are seen in Figure 18. We see that the trained Topoformer provides superior neural fits, and that language network voxels are predicted better than the control brain regions. This confirms the previous results found using the PLS-SVD alignment approach. We note that the relatively low performance of the encoding model is partly attributed to the fact that we included a large subset of voxels. The results obtained via PLS-SVD in Figures 5 and 15 indicate that the shared variability *across* voxels aids the alignment between brain and model. Finally, we note that although the untrained Topoformer-BERT had lower predictivity performance than the trained counterpart, it was still above zero, in line with prior findings (Schrimpf et al., 2021; Caucheteux & King, 2022; Pasquiou et al., 2022).

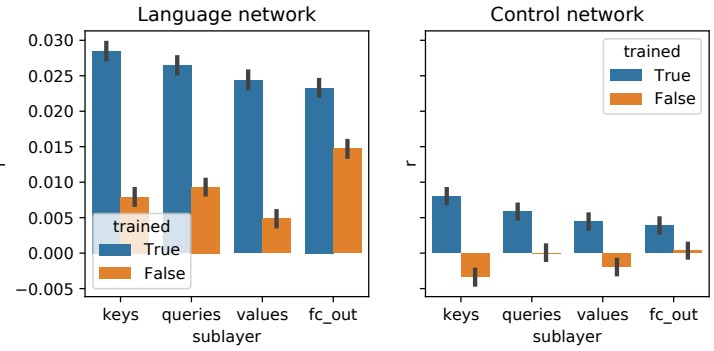

**Figure 18:** Encoding model prediction results across two control analyses: Comparison between the language network versus non-linguistic (motor-related) brain regions, and comparison of the trained versus untrained Topoformer-BERT model.

