# OpenReview forum: "Topoformer: brain-like topographic organization in Transformer language models through spatial querying and reweighting"
_ICLR.cc/2024/Workshop/Re-Align — ICLR 2024 Workshop Re-Align Poster_

### Official Review · Reviewer_4mf1 · 2024-02-24
**A solid start to creating more brain-like and interpretable LMs but the insights presently obtained from this are a little lacking**

**Rating:** 2
**Fit:** 3
**Confidence:** 2

**Workshop Review:**

In this work the authors seek to embed Language models with topographical organization similar to the type of topological organization seen in many areas of the brain. To achieve this the authors add two constraints to the traditional attention layer of their transformer architecture: a spatial querying term that pools together local queries within some prespecified receptive field and spatial reweighting that locally weights the attention values. The authors very clearly show that these two constraints leads to topographical activity in their Topoformer network and show that it only slightly reduces performance. Additionally the authors design 8 datasets specifically designed to test sensitivity to distinct linguistic properties. Surprisingly, they show that the constraints they added simply to impose topography onto the network also lead to interpretable functional organization. This finding is noteworthy, but I do wish that it was followed with ablation experiments to test how modular their network is as well. Additionally, here the topography seems an inconvenience to the network (as performance always decreases when adding the two constraints) but I imagine that there are situations where it would actually benefit the network as well. For example, the induced topography may make the model more robust to noise in the token as both the queries and values are being pooled now. This would potentially help in answering the question of if the topography we see in the brain is simply a by product of biological constraints that evolution never found a way to relax or if these constraints endow the brain with a functional advantage as well.

 After showing that they can successfully induce topographical structure into their transformer,  the authors ask how similar their model behaves to the brain. To do this they collect fMRI data from 6 participants and compare model response to fMRI. Their analyses argue that Topoformer is aligned with human data but to show this they develop a novel method making it difficult to interpret the magnitude of the alignment. Here it would have been nice to see what typical alignment scores are between the different subjects and maybe using this as an upper bound. Importantly, the authors do perform a negative control comparing alignment between Topoformer and motor-related activity and show that there is no alignment: suggesting that their metric is not so powerful that it always finds alignment. Additionally, the authors do also calculate alignment using more traditional encoding metrics, but I found the results here puzzling. Specifically, while they found that the topoformer had an above zero prediction score: their largest r was 0.03 this is almost an order of magnitude lower than alignments reported in previous literature that had a minimum alignment of around 0.2. Here again it would help to have a ceiling for how high these alignments could possibly be in their dataset. Further, it’s unclear how much the induced topography aids in model alignment. The authors should run a control using a performance matched BERT model to see whether their topography actually increases brain alignment more than just having more accurate models does.

All that aside, the work presented here as is already very promising and presented very clearly, but has the potential to be much better with some more careful comparisons to the brain.

**Reason For Not Giving Higher Score:**

My main reason for not giving this a higher score is that I don't know how to think about their alignment results. The results shown in the paper are greater than zero but when they use previously used methods for quantifying alignment they get scores an order of magnitude lower than previous reports. This could all be explained if their subjects also had low alignment between them, but without that information it's hard to interpret.

I also wish that they explored certain avenues more but those are really just points of curiosity.

**Reason For Not Giving Lower Score:**

From an ML perspective this paper is quite strong. They develop two constraints to the traditional attention layer that leads to topographical activity and functional selectivity in a LM. To my knowledge this is the first time anyone has ever demonstrated either of these two points.

**Reviewer Domain:**

neuroscience

---

### Official Review · Reviewer_BSi7 · 2024-02-28
**Topoformer: brain-like topographic organization in Transformer language models through spatial querying and reweighting**

**Rating:** 3
**Fit:** 3
**Confidence:** 2

**Workshop Review:**

**Summary:**

DNNs typically have no topographic organization in the layers or spatial priors while these types of organization are key to neural computations. Successful NN models are not often interpretable. The authors provide a method of providing topographic organization in the layers of a Transformer network, forming so-called ‘Topoformers’.
This is done through two adjustments.

*Spatial querying (SQ)*: A binary intermediate matrix $M$ is inserted into the attention computation, associating local pools of queries with a single key. Locality is parameterized with parameter $r_{SQ}$. Representations of keys will have spatial correspondence with queries.

*Spatial reweighting (SR)*: Converts the fully-connected layer into a locally connected layer by using a locally connected re-weighting matrix $W^O_{local}$ with parameter $r_{SR}$. Large positive weights were used to encourage topographic organization.

Topoformer networks perform similar to, often slightly worse than non-topographic control models on NLP benchmarks, while retaining interpretable topographic organization. Using fMRI analysis of human brain responses to naturalistic sentences, they show alignment between the variability in the Topoformer and the language network in the brain.

*Experiment 1*: 1-layer single-head encoder-only Topoformer on the IMDB sentiment analysis dataset. Transformer without SQ or SR had accuracy of 0.83. Topoformer with only SQ on achiveved 0.81, with SR achieved 0.75. Topoformer-SQ produced topography in keys and queries,  which Topoformer-SQR also had topography in the values and output layers.

*Experiment 2*: BERT model using a Masked Language Modeling objective on the Bookcorpus-Wikipedia dataset. Using the GLUE benchmark, the Topoformer performance was similar to the non-topographic counterpart. Topographic organization was quantified using the generic topography $t_g$ statistic, which averages correlation over a range of scales. To interpret the organization, they evaluated the selectivity of activations to eight test suites containing linguistic or semantic properties such as Intactness, Animacy etc. While the unit activations showed selectivity towards contrasts, these were different across contrasts, implying that distinctions between the properties were represented in the topographic activity pattern differences. Selectivity was based on distinctions from the overall topographic activity pattern. With minimally different sentences, the effects were lower and the Topoformer showed weak selectivity to sentences with correct syntactic agreement.

*Experiment 3*: Recorded brain responses from 5 participants reading 1000, 6-word sentences, some of which were non-word strings. Voxels with preference for sentences were chosen as the ‘language network’. Computed $t_g$ on unsmoothed BOLD responses in these voxels, which indicated topographic organization.

Using PLS-SVD, they performed representational alignment between the language network response and Topoformer-BERT layers. In general, the first two components were strongly aligned for each sublayer. Low-dimensional variability between activity in the brain’s language model and the Topoformer can thus be aligned.

**Comments:**

Overall, explicitly introducing topography into an NN model is a good way of increasing interpretability. However, as the authors found out and discuss, this does not guarantee interpretability as there can still be significant variability and polysemanticity even with topographic organization. It is unfortunate that the Topoformers showed no marked performance improvements – however this may be different in tasks that explicitly demand spatial reasoning and separation. The analysis and alignment to human fMRI data is a challenging but well-explained and well-executed work that greatly increases the applicability and impact of this paper. The authors have done an admirable job of explaining their methodology and rationale in the appendix.

It is not clear why the $t_g$ statistic was not employed for the first test on the IMDB sentiment analysis dataset. I believe this should be done, as it is good to quantify the organization of the layers in this analysis.

Similarly, it is not clear why the SQ and SQR versions were not tested for Topoformer-BERT.

*Sec 3.1*: “The local connectivity in the spatial querying and spatial reweighting operations are controlled through a hyperparameter $r_{SR}$ that sets the radius of spatial receptive fields (RFs).” I believe this should be hyperparameters $r_{SQ}$ and $r_{SR}$.

*Figure 3*: It may be a lot clearer if the statistics in A are shown not as a bar graph, but as datapoints like in B.

General comment for all figures: please increase the font size in the axes labels and especially the legends.

**Reason For Not Giving Higher Score:**

N/A

**Reason For Not Giving Lower Score:**

The paper is an excellent fit to the workshop as it directly deals with creating representations in a machine learning model and aligning it with biological represenations. The work is complex, yet clearly explained and well-analyzed.

**Reviewer Domain:**

neuroscience

---

### Official Review · Reviewer_ryoZ · 2024-02-28
**The paper shows potential in tackling an important problem to representational alignment, in the context of natural language processing.**

**Rating:** 3
**Fit:** 3
**Confidence:** 2

**Workshop Review:**

This work is concerned with the lack of spatial constraints in machine learning models, in particular for the transformer architecture, which hinders comparisons with the brain. The paper aims to implement such spatial constraints in the transformer, by making changes that encourage topographic organization to the self-attention layer. The more biologically plausible topographic structure may closer resemble structure found in the brain, and potentially be easier to interpret.

The paper aims to provide evidence using natural language experiments that
 - These changes result in the transformer learning a topographic structure.
 - The emergent topographic structure has hints of increased intepretability.
 - The structure shares some topographic alignment with brain responses.


Strong points:

The approach of adding topographic organization to transformers comes across as important and relevant in the context of representational alignment between machine learning models and the brain, for the following two reasons:

It seems likely that the existence of spatial constraints will affect the representational structure that is formed in the learning process. Thus, in order make comparisons between machine learning models and the brain in the area of natural language processing, we should consider using a spatially constrained architecture, which the leading NLP architecture, the transformer, currently is not.

Additionally, it is plausible that topographic organization aids in interpretability. If so, this may help in crafting comparisons between different representational structures.

The methods used to implement topographic organization in the transformer architecture make sense and are relatively straightforward. It is clear why they may result in topographic organization and convincing experimental evidence is provided for this as well.

Results hinting at increased interpretability are interesting and show potential. Similarly, results for the alignment displayed in between topoformers and the brain hold promise.

Experimental detail were fairly comprehensive, and for the most part relevant control experiments and additional experimental details were present, or could be found in the appendix.


Weak points:

This work could benefit from further experimental work. The addition of either

 - an experiment more explicitly demonstrating how topographically organized transformers can be seen as more interpretable
 - an experiment providing more evidence of alignment between the brain and the topoformer

would add a lot, as the experiments currently presented still feel a little limited in scope to tackle the questions raised. However, the topoformer approach appears sound and holds potential.

Although methods were mostly well-detailed, there were a few occasions were some additional clarity could be added.

For instance, in section 2.2 the process of converting the outer reweighting matrix $W^O$ to a locally connected matrix $W_{local}^O$ is not made very formally explicit. An example or an explicit formula may add some clarity.

Another example would be in figure 3, where it would be nice to see a typical value of the topographic statistic for the non-topographic control model. This gives us a frame of reference to understand the degree of topography displayed here in topoformer-BERT, as this topographic statistic I believe is not common knowledge.



I recommend this work to be accepted, the main reason being that I find it convincing that the lack of topography in transformers is important for representational alignment in context of natural language processing, and should be further discussed. The approach taken here makes sense, the methods are sound, and there is potential for interesting results.


Some additional questions:

The motivation to restrict to a single head is clear, but I wonder if this is not a limitation, as in practice often many heads are needed to capture different dependencies. Will this limit the topoformer when scaling to larger and more complex language datasets? Perhaps you could elaborate on why this is/isn't important, or how to remedy it.

For the PLS-SVD comparison between the topoformer and the fMRI data there is a control experiment with an untrained topoformer. Perhaps a control experiment with the trained non-topographic transformer can also be valuable, to determine the importance of the topographic constraints introduced here in the topoformer architecture to the alignment.

I find the (somewhat large) differences in selectivity patterns for the different constraints very interesting. It would be interesting to know where they are coming from, e.g. is there some property of the data in the test suites that is correlated to the type of pattern you get?


Minor comments:

The text in some plots is pretty small and hard to read.

**Reason For Not Giving Higher Score:**

N/A

**Reason For Not Giving Lower Score:**

The approach of adding topographic constraints to the transformer architecture seems like a promising path towards studying alignment between language processing areas in the brain and transformer architectures. The paper takes steps in this direction I think it is worth additional exploration and discussion.

**Reviewer Domain:**

machine learning

---

### Decision · Program_Chairs · 2024-03-02

Accept (Poster)